# Associations between Autistic and ADHD Traits and the Well-Being and Mental Health of University Students

**DOI:** 10.3390/healthcare12010014

**Published:** 2023-12-20

**Authors:** Japnoor Garcha, Andrew P. Smith

**Affiliations:** Centre for Occupational and Health Psychology, School of Psychology, Cardiff University, Cardiff CF10 3AS, UK; garchaj@cardiff.ac.uk

**Keywords:** autism, ADHD, autistic traits, mental health, well-being, strengths and difficulties, well-being process questionnaire

## Abstract

Research on autism and ADHD continues to increase, as does the research on well-being and mental health. There is a growing need to understand what factors impact mental health and well-being, and the question arises as to what factors impact mental health and well-being in autism and ADHD. The existing literature focuses on two different aspects when it comes to the well-being and mental health of autism in students. One aspect focuses on mental health and well-being in diagnosed neurodivergent individuals, and the other aspect focuses on associations between autistic and ADHD traits and mental health and well-being. In order to understand the impact on mental health in autism, an online survey using the Qualtrics platform was given to a sample of 430 university students. The survey used the well-being process questionnaire, the autism spectrum quotient, the ADHD self-report scale, and the strengths and difficulties questionnaire. The results showed significant correlations between anxiety, depression, and autistic and ADHD traits (all correlations > 0.2). These variables were also correlated with the well-being and SDQ outcomes and well-being predictors (all correlations > 0.2). The regression analyses showed significant associations between well-being outcomes and predictor variables and anxiety and depression, whereas the effects of autistic and ADHD traits were restricted to the SDQ outcomes (hyperactivity, conduct, and peer problems). Regression analyses were also conducted to determine whether a variable formed by combining autistic traits, ADHD, anxiety, and depression scores was a significant predictor of well-being and SDQ outcomes. The combined variable was associated with all outcome variables except the prosocial variable. This study provides the basis for further research for understanding the interaction between well-being, mental health, autism, and ADHD.

## 1. Introduction

Mental health issues in university students are an increasing topic of concern. University students are facing a mental health care crisis. Autism and ADHD have been recognized as primary special education needs. Our previous study [1] aimed to understand the link between autism and ADHD traits and well-being and mental health using the well-being process questionnaire (WPQ) and the strengths and difficulties questionnaire (SDQ). There is a need to use a holistic approach [2] to understand the link between mental health and autistic and ADHD traits. The focus in previous research studies has been on specific aspects of either positive well-being or negative well-being. The holistic focus is meant to look at all aspects of well-being and not focus on only one aspect. Most studies in the autism literature have not used a holistic approach. This study is the first to do so with university students. This study was based on a previous study conducted to understand the associations between autism and ADHD traits and secondary school students’ well-being. Mental health and autism have been strongly associated, and the literature on this and well-being is considered in the next section.

### 1.1. The Literature on Autism, Well-Being, and Mental Health

In order to understand the link between mental health and autism, it is first essential to understand what precisely mental health entails and how it is defined. Mental health is often divided into mental health problems and mental health. Keyes [3] identified three components of mental health: emotional well-being, psychological well-being, and social well-being. After definitions did not show a complete picture, a lot of research and studies later formulated a new definition [4], which is often referred to as a definition of mental health:

“Mental health is a dynamic state of internal equilibrium which enables individuals to use their abilities in harmony with universal values of society. Basic cognitive and social skills; ability to recognize, express and modulate one’s own emotions, as well as empathize with others; flexibility and ability to cope with adverse life events and function in social roles; and harmonious relationship between body and mind represent important components of mental health which contribute, to varying degrees, to the state of internal equilibrium”.

Well-being is a broad term, and over the years, there have been numerous attempts to define it. These attempts have helped shift the focus from well-being only considered in terms of happiness and being free from sorrow to a more holistic outlook, including good mental health, life satisfaction, and coping well with stress.

Previous research [5] found that ASD symptoms influence well-being and are funnelled by factors like feelings of safety in one’s environment and psychological satisfaction. ASD symptom severity has also been linked to psychological well-being. It has been widely accepted that subclinical autistic traits are distributed continuously in the general population. Research [6] found that children with ASD scored lower than typically developing children on “subjective well-being.” A research study [7] also found a significant association between autistic traits and mental well-being. Positive well-being is the cognitive and affective reactions to the world, including social integration and positive progress in time. It includes life satisfaction, positive mood, and energy as components. Negative well-being can be defined as the cognitive and affective responses to perceived deficits. Distress, negative mood, symptoms, and hyperarousal are some components of negative well-being.

Recent research has also suggested that how one copes with stress is associated with one’s mental health and well-being. Research studies have suggested that losses in mental health result in increasing odds of mental illness over time. According to the World Health Organization (WHO), mental health is “a state of well-being in which the individual realizes his or her abilities, can cope with the normal stresses of life, can work productively and fruitfully, and can contribute to his or her community” [8].

A search yielded close to 8000 studies on PubMed for autism and mental health. A meta-analysis [9] of 96 studies found that the overall prevalence of mental health problems associated with autism was 28% for ADHD, 20% for anxiety disorders, 13% for sleep–wake disorders, 12% for disruptive impulse control and conduct disorders, 5% for bipolar disorders, and 4% for schizophrenia spectrum. An umbrella review of 12 meta-analyses [10] found that there was a high level of comorbid psychiatric disorders in people with Autistic Spectrum Disorder (ASD). These include anxiety disorders, mood disorders, depressive disorders, ADHD, and conduct and impulse control disorders. Research [11] has found that 50% of adults with ASD met the diagnostic criteria for Social Anxiety Disorder (SAD). Studies have highlighted the importance of community-based studies and an inclusive sample, as age, gender, intellectual ability, and country of study all add to heterogeneity. It was found [12] that the current and lifetime prevalence of anxiety disorders was between 27% and 42% for adults with ASD and between 23% and 37% for depression in adults with ASD. These include anxiety disorders, mood disorders, depressive disorders, ADHD, and conduct and impulse control disorders. The previous research has also shown that children with high-functioning ASD are at a greater risk of developing anxiety symptoms. It has been seen that disengagement coping is associated with poorer mental health in the autistic population [13]. The literature shows that adults with autism spectrum disorders are at a higher risk of developing comorbid mental health problems, with anxiety and depression at the forefront of these conditions. The other literature also shows that anxiety and depression are significantly related to ASD symptomology [14].

### 1.2. Extension of Previous Research to University Students

The previous research on the associations between ADHD and autistic traits and well-being in secondary school students identified particular associations. It is essential to determine whether this applies to other age groups, and the present study examined this topic in a sample of university students. A PubMed search for autism in university students returned close to 1600 results. The search on autism in children and adolescents yielded close to 15,000 search results and studies. A systematic review [15] of factors impacting mental health in university students found that those with autism were more likely to develop and experience mental health issues. Colleges and universities all over the world have seen a significant increase in students with autism and ADHD, and further research in this context is required. The previous study did not include measures of mental health (e.g., anxiety and depression). A brief overview of the previous study is given in the next section.

### 1.3. An Overview—Association between Autism and ADHD Traits and Well-Being in Secondary School Students

The study conducted in 2022 in a post-COVID environment was the first to apply the well-being process framework to understand well-being, autism, and ADHD holistically. The study used the short-form WPQ and SDQ to examine associations between autism and ADHD traits and the well-being of secondary school students. 

The well-being process questionnaire did not just measure subjective well-being outcomes (e.g., happiness, life satisfaction, and positive affect) but included adverse outcomes (perceived stress, negative affect, and mental health problems). The well-being process model also included variables that predicted well-being outcomes. These included negative factors such as exposure to stressors and negative coping (avoidance, self-blame, and wishful thinking). Positive influences such as social support, psychological capital (self-esteem, optimism, and self-efficacy), and positive coping (problem-solving and use of social support) were also included. Previous results showed that positive well-being is strongly associated with positive factors and, to a lesser extent, the absence of negative factors. The opposite profile of associations was found with negative well-being.

Autism and ADHD are both neurodevelopmental conditions, and research has found they co-occur, with the rate of co-occurrence varying between 14% and 78%. They also share similar impairments and common structural brain abnormalities. Hence, the study included both autism and ADHD, as there is an overlap between the two. The previous literature shows that many studies have used the well-being process questionnaire with students, and these research studies have replicated the effects of established predictors and added new predictors (e.g., workload, work-life balance, daytime sleepiness, flow) and outcomes (e.g., flourishing; physical health). In the case of autism and ADHD, it was apparent that other outcomes needed to be considered. Hence, the SDQ was used for this purpose. The SDQ is a behavioural screening questionnaire developed for children and adolescents. It is used to measure emotional and behavioural difficulties and prosocial behaviour. The domains covered in the SDQ are susceptible to autism and ADHD. Prosocial behaviour and peer relationships are the most significant outcomes as they take up a considerable part of the lives of children with autism and ADHD, and problems may continue into adolescence and adulthood.

The correlational analyses showed a significant association between autistic and ADHD traits. Autistic and ADHD traits were significantly correlated with established predictors of well-being and well-being outcomes. Regression analyses with both autistic and ADHD traits as predictors showed that associations with well-being outcomes were primarily with ADHD, with some overlap with autistic traits. Other regressions confirmed previous research studies and showed that positive well-being outcomes were positively associated with positive predictors like psychological capital and negative outcomes with negative predictors like negative coping and stressful experiences. 

The analyses then included autistic and ADHD traits and established well-being predictors to the regression model. These analyses revealed no significant effects of autistic and ADHD traits on well-being outcomes. There were, however, significant associations between SDQ outcomes (reduced prosocial behaviour and hyperactivity) and autistic and ADHD traits. 

### 1.4. Impact of Mental Health Traits and Autism

There have been seven reviews on this topic, and all of these have the same conclusion that poor mental health is frequently reported with a higher prevalence of anxiety and depression among autistic adults compared with the general population. Another study [16] also found that anxiety and depression are the major comorbid problems in children and young people with autism spectrum disorders. Research shows that co-occurring mental health conditions are more prevalent in the autism population than in the general population. Most research studies focused on various mental health disorders have found a link or co-occurrence of anxiety and depression with autism. Studies have also found anxiety and depression to be highly correlated with each other and also highly correlated with the severity of autism [17]. A meta-analysis of 96 research [9] studies found that co-occurring anxiety in individuals with autism amplifies autistic symptoms, including social impairments, sensory features, and repetitive behaviours [18,19], and might be associated with the development of depression [20], contributing to an increased risk of suicide and early mortality. The co-occurring mental health conditions in autism tend to persist from childhood to adolescence. There is also evidence [21] that suggests the prevalence of co-occurring psychopathology increases in adults with autism. Studies [22] have found a high prevalence of co-occurring mental health conditions in the autism population, which is most of the time also significantly higher than in the general population. Mental health problems are increasingly becoming common these days. A 2014 survey (NHS) [23] on mental health and well-being found that 1 in 16 people over the age of 16 have a mental health problem. There has been a substantial increase in the prevalence since 1993. In 2023, the CDC [24] reported that that 1 in 36 children now have autism. The current estimates from 2018 [25] suggest that the autism prevalence in the UK (United Kingdom) is 2%. The reported global prevalence of ADHD is 5%. The prevalence of neurodevelopmental disorders is increasing exponentially as well.

Most studies conducted on autism and well-being have lacked a holistic approach. There have been limited studies that have in any way tried to study the impact of well-being on mental health in students with autism traits. A search using PubMed generated limited research studies (close to eight) on the link between well-being and mental health in autistic students. The studies conducted have seen a minute focus on the mental health of university students, but no visible attempt has been made to understand the interaction and impact of mental health and well-being in university students with autistic traits. To bridge this gap, the current study on university students has opened the door to further understanding and developing research in this area. 

### 1.5. Will Mental Health Variables Combined with Autism and ADHD Be Stronger Predictors of Outcomes as Compared with Individual Variables?

A study on the holistic approach to well-being in nurses [2] used the WPQ to determine if combined predictors strongly indicate well-being. The study was conducted on 177 nurses. The original baseline study conducted in 2017 used individual predictors and well-being scores. However, the study also used a combined predictor score as it was hypothesized this would have the highest correlation with the outcomes. This research also demonstrated that using a short form of the WPQ to measure well-being is possible, and combined effects as a measure can be derived simply by using all the predictor variables. 

The present study continued the research examining the extent to which autistic and ADHD traits are associated with well-being and strengths and difficulties. In addition, it investigated the extent to which trait anxiety and depression were associated with these outcomes. Following the analysis of the individual effects of these traits, analyses were conducted to determine whether combinations of traits had additive effects on the well-being and SDQ outcomes.

### 1.6. Gaps in Our Knowledge of These Topics

Most studies conducted on autism and well-being have lacked a holistic approach. There have been limited studies that have in any way tried to study the impact of well-being on mental health in students with autism. A search using PubMed generated limited research studies for the link between autistic, ADHD, anxiety and depression traits and the well-being of university students. The studies conducted focused on the mental health of university students, but no visible attempt was made to understand the interaction and impact of mental health and well-being in university students with autistic traits. To bridge this gap, the current study on university students was designed to help understand and develop research in this area further. The specific hypotheses tested are given in the next section.

### 1.7. Aims and Objectives

The first aim was to examine associations between autistic and ADHD traits and well-being and strengths and difficulties of university students.The second aim was to examine the associations between well-being, strengths and difficulties, and mental health traits.The next aim was to conduct analyses controlling for established correlates of well-being and examining whether the associations between autism, ADHD, anxiety and depression traits and well-being and strengths and difficulties remain consistent and significant.The final aim was to examine the relationship between the combined variable, the well-being process, and the SDQ variables.

## 2. Methods

### 2.1. Participants and Sample Size

The participants were all students taking psychology courses at Cardiff University in South Wales. Four hundred and thirty students (84.6% female; evenly distributed across first, second and third/MSc years) took part in this study and were given course credits for their participation. Students enrolled in these courses are predominantly female, and this was reflected in the present sample.

### 2.2. Materials

It has been seen that as a neurodevelopmental disorder, autism is characterized by various signs and symptoms often referred to as traits. These traits broadly fall under two categories—social communication and interaction skills and restricted or repetitive behaviours. Traits like delayed language, delayed cognition, hyperactivity, and impulsive or inattentive behaviour, to name a few, do not fall under the two categories mentioned and are grouped as other traits. Sometimes, it is hard to differentiate these traits from similar traits such as ADHD or mental health problems like anxiety and depression. Hence, to measure autistic traits specifically and precisely, the autism spectrum quotient (AQ) was used. 

As discussed in the Introduction, well-being is a complex construct, and it is crucial to understand the essence of the term and everything it entails. The area is being increasingly researched, and there is now an increased demand to extend the study beyond disorders, disabilities, and deficits. After going through the literature and keeping this in mind, the current study was designed. The well-being process model described in the Introduction was used in our previous study with secondary school children. It was used here because it includes variables influencing well-being outcomes and covers both positive and negative domains. 

#### 2.2.1. Questionnaires

An online survey using the Qualtrics platform was carried out to understand the associations between autistic and ADHD traits, well-being, and mental health. The dependent variables in this study were positive well-being, negative well-being, physical health, flourishing, emotional problems, hyperactivity, conduct, peer relationships, and prosocial behaviour. The predictor variables were the total scores of AQ-10 and the sub-score from part A of the ADHD self-report scale, work–life balance, social support, student stressors, sleepiness, positive coping, negative coping, psychological capital, flow, and rumination. The survey also used anxiety and depression as both predictor and outcome variables. Details of the questionnaires are given in the next section. The survey consisted of 4 different questionnaires, which were necessary for this research. 

##### AQ-10

The autism spectrum quotient is a questionnaire designed to measure the expression of autism traits in an individual based on his/her self-assessment. It was initially a 50-item questionnaire, but shorter versions have been created. A 10-item [26] scale was used here. It consists of a 4-point Likert scale ranging from “definitely agree” to “definitely disagree”. Scores range from 0 to 10, and total AQ scores were used in the analyses. 

##### ADHQ

The ADHD self-report scale, the ASRS, was devised in collaboration with the World Health Organization. It has been used as a diagnostic tool [27] and includes 18 questions. The first part of the scale has six questions that stand out in terms of being the most predictive of ADHD symptoms. It uses a five-point Likert scale. The participants were asked to answer each question by rating on the Likert scale ranging from never to very often. Part A was used here, and the scores ranged from 0 o 6. The total ADHD scores were used in the analyses.

##### Strengths and Difficulties Questionnaire (SDQ)

The strengths and difficulties questionnaire is an abnormal behaviour screening questionnaire. It comprises 25 items [28] spread over five subscales—emotional symptoms, conduct problems, hyperactivity/inattention, peer relationship problems, and prosocial behaviour. The items are based on a 3-point Likert scale ranging from untrue to undoubtedly true. Responses are dichotomized, and then the items in each subscale are added to give the score for that scale. The scale scores were used in the present analyses.

##### Short-Form Student WPQ

The present short-form well-being process questionnaire [1] was adapted from the original WPQ [29]. The predictor variables were student stressors, negative coping, workload, work–life balance, daytime sleepiness, psychological capital, social support, positive coping, and flow. The dependent variables were positive well-being, negative well-being, physical health, and the extent to which the person was flourishing. The short form of the WPQ also includes single-item questions about anxiety and depression, which are highly correlated with the scores from the Hospital Anxiety and Depression (HADS) questionnaire. 

### 2.3. Combined Effects

Previous research was conducted to determine if the short-form WPQ could be used to determine if combined predictors strongly indicate well-being. This knowledge, combined with the previous study conducted with secondary school students, led to the hypothesis that the combination of traits will produce additive effects on well-being and SDQ outcomes. Keeping this in mind, the combined variable was generated as the sum of quartile scores of various variables: AQ, ADHD, anxiety, and depression.

### 2.4. Procedure

This study was conducted using the Qualtrics platform. The survey included an informed consent form at the beginning that contained details, and it explained to the students that they could choose not to continue if they felt uncomfortable and could refuse to answer specific questions. It also explained that the data would be anonymous. This study was linked to the Cardiff University EMS (Experimental Management System) system, participants were recruited through EMS, and course credits were given upon completion of the survey.

### 2.5. Analysis Strategy

This study was conducted with the use of different sets of analyses that enabled an understanding of the associations between variables. SPSS version 27 was used to conduct all analyses. The initial analysis included descriptive statistics for all variables. Pearson’s correlation analysis was used to examine the association between autism and ADHD scores as well as the correlation between these scores and predictor and outcome variables. A series of linear regression analyses were also carried out. The first set of regression included AQ and ADHD as predictors. The second set included the predictors from the well-being process questionnaire. The third set included AQ, ADHD, and established predictors from the WPQ. In the final set of regressions, there was a new combined variable and the established predictors from WPQ to determine the association between AQ, ADHD, well-being, and mental health.

## 3. Results

### 3.1. Descriptive Statistics

The descriptive statistics for both the outcome as well as predictor variables are shown in Table 1. There was considerable variation in each measure. The scores were comparable to previous research findings, and there was very little missing data. The missing data consisted of six participants who had left one or two questions blank. There were two participants that left a considerable amount of the questionnaire blank, and they were not included in the analysis. It was found that 30 participants fell on or above the high threshold cut-off, where a score of 6 or above meant they warranted an autism diagnostic assessment. 

### 3.2. Association between AQ, ADHD, Anxiety and Depression Scores, and Outcome Variables

The correlations between AQ, ADHD, anxiety, depression, and the outcome variables are shown in Table 2. The outcome variables included variables from the WPQ, namely, positive and negative well-being, physical health, flourishing, and the SDQ subscales of conduct, hyperactivity, emotional, peer, and prosocial behaviour. As expected, the anxiety and depression scores were significantly correlated and were correlated with the AQ and ADHD scores. They were also significantly correlated with the majority of the well-being outcomes. AQ and ADHD were significantly correlated with all the well-being outcome variables except prosocial behaviour.

### 3.3. Association between AQ, ADHD, Anxiety, Depression, and the Predictors of Well-Being

These correlations are shown in Table 3, which also has the established predictors of well-being, which are variables on the WPQ that have repeatedly been shown to be associated with the outcomes in the previous literature. Anxiety and depression were significantly correlated with most of the well-being predictors, as were AQ and ADHD. Depression was significantly correlated with all variables, and anxiety was significantly correlated with all except flow and rumination. AQ and ADHD were significantly correlated with all variables except rumination. In the correlation analysis, each IV was selected against DV to see the correlation or internal associations these variables have with each other.

### 3.4. Regressions with AQ, ADHD, Anxiety, and Depression as the Predictor Variables 

The initial analyses aimed to examine associations before the established predictors were covaried. Separate regressions were carried out for each outcome and established predictor variable. The results are shown in Table 4. The well-being outcomes (positive well-being, negative well-being, and flourishing) were predicted by anxiety and depression. Physical health was predicted by anxiety and depression and also AQ. Apart from conduct problems, depression also predicted the SDQ scores. AQ and/or ADHD scores were associated with conduct problems, emotional problems, hyperactivity, peer problems and prosocial behaviour. The next set of regressions included the established predictors of well-being in the regression model.

### 3.5. Regressions with AQ, ADHD, Anxiety, Depression, and Established WPQ Predictors 

Separate regressions were carried out for each outcome, and the results of these are shown in Table 5 below. The established predictors of the WPQ have been generated based on previous research studies that used the WPQ. Depression was significantly associated with the outcomes apart from physical health, flourishing, and hyperactivity. Anxiety was significantly associated with the well-being outcomes. AQ and/or ADHD were only associated with the SDQ outcomes of hyperactivity, conduct, and peer problems. Positive well-being was associated with social support and positive pondering. Negative well-being was associated with stressors and rumination. Good physical health was associated with a low workload and positive pondering. Flourishing or thriving was associated with high psychological capital and flow. Lack of social support was also associated with conduct and peer problems.

### 3.6. Combined Variables and the Associations between Predictor and Outcome Variables

A number of variables were correlated, such as AQ, ADHD, anxiety, and depression. The correlation between these variables was not highly significant but was significant to a certain extent. In such a situation, it is beneficial to look at the combined effect of these variables. In order to understand this better, the quartile score for each variable was calculated and combined to form a new variable called the “combined variable”. A series of analyses were run with the combined variable to examine the associations with predictor and outcome variables.

#### 3.6.1. Correlation Analysis of the Combined Variable with Outcomes and Predictors

The correlation analysis showed that the combined variable was significantly correlated with all outcome variables except the prosocial variable (see Table 6). The combined variable was found to be positively correlated with negative well-being, conduct scores, emotional scores, hyperactivity scores, and peer problem scores. The combined variable was found to be negatively correlated with prosocial behaviour score, flourishing, physical health, and positive well-being.

The second set of correlation analyses showed that the combined variable was significantly correlated with all predictor variables except rumination (see Table 7). The combined variable was found to be positively correlated with anxiety, depression, AQ, ADHD, stress, negative coping, work–life balance, workload, and sleepiness. The combined variable was found to be negatively correlated with flow, psychological capital, positive coping, and social support. 

#### 3.6.2. Regression Analysis of the Combined Variable and Established Predictors

The regression analysis that was conducted showed that positive well-being was significantly associated with the combined variable and psychological capital (see Table 8). It was seen that negative well-being was significantly associated with the combined variable and stressors. Physical health was significantly associated with the combined variable only. The outcome variable of flourishing was significantly associated with stressors, psychological capital, and flow. It was seen that conduct problems were significantly associated with only the combined variable. Emotional problems were significantly associated with the combined variable, negative coping, psychological capital, and sleepiness. 

## 4. Discussion

The present study examined the association between autistic and ADHD traits, mental health, and well-being in university students. There has been research focusing on children and young adults with a diagnosis of autism or ADHD. There are, however, also people who have these characteristics linked with autism and ADHD who have not been given a formal diagnosis. The fast-paced world of today has deep impacts on individuals, especially when there is change taking place, like going to university and living away from home. Mental health is a topic of increasing importance, and there are individuals who have had formal diagnoses of anxiety and depression, and also those with high trait anxiety and depression levels. The present study examined the individual and combined effects of autistic, ADHD, anxiety, and depression traits on well-being and SDQ outcomes.

The previous study with secondary school students exhibited a series of results that gave insight into the associations between AQ and ADHD. The univariate analysis of outcome and predictor variables showed that AQ and ADHD were significantly correlated. They were also significantly correlated with many outcome variables. It also showed that those students who had high autistic and ADHD trait scores had lower levels of well-being. However, AQ and ADHD scores were also significantly correlated with established predictors of well-being. A series of regression analyses showed varying results. When AQ and ADHD were added in the same regression analysis as predictors, it was seen that there were associations between ADHD traits and all outcome variables except the prosocial variable, whereas AQ only had associations with hyperactivity, peer problems, and prosocial variables. A previous study [30] conducted in Norway found that adults with a dual diagnosis of ADHD and autism were more likely to experience co-occurring mental health disorders, commonly, anxiety disorders.

The initial analysis conducted in the present study revealed that autistic and ADHD traits were associated with well-being and SDQ outcomes. The next linear regression analysis including AQ, ADHD, and the established predictors of well-being in the model showed that there were no significant effects of AQ and ADHD on well-being outcomes. The results from the previous study were, therefore, replicated to some extent with the sample of university students. 

The initial analysis confirmed that autistic and ADHD traits are correlated to significant degrees. The analysis also showed that AQ and ADHD were significantly correlated with various outcome variables when both AQ and ADHD were included in the same set of analyses. However, AQ had no association with prosocial behaviour and flourishing. The analysis also found that ADHD scores were correlated with all outcomes except prosocial behaviour. The present study extended our previous research by including trait anxiety and depression in the analyses. The autistic and ADHD traits were also associated with anxiety and depression. Anxiety and depression served as both outcome and predictor variables. It was seen that in the correlational analysis, anxiety was significantly correlated with depression, AQ, ADHD, conduct, hyperactivity, emotional, peer, positive well-being, negative well-being, physical health, and flourishing. It was also found that depression was significantly correlated with anxiety, AQ, ADHD, SDQ, conduct, hyperactivity, emotional, peer, prosocial, positive well-being, negative well-being, physical health, and flourishing. On further analysis with AQ, ADHD, and established predictors, it was found that there were some associations that remained significant. It was found that positive well-being and negative well-being, flourishing, and emotional problems were associated with both anxiety and depression. Physical health, prosocial, peer problems and hyperactivity were also associated with depression. Conduct, hyperactivity, emotional and prosocial behaviour were also associated with AQ and ADHD. Apart from these, the analysis also showed that physical health and peer problems also had associations with AQ. It was also seen that when anxiety and depression were added as predictor variables, stressors were associated with anxiety and depression, whereas social support and positive coping were significantly associated with depression and AQ. Negative coping was associated with anxiety, depression, and ADHD. It was found that work-life balance and sleepiness were associated with anxiety, depression, and ADHD. Flow was associated with ADHD. Psychological capital, positive coping and social support were associated with AQ. 

Further analysis of the combined variable (AQ, ADHD, anxiety, and depression combined) also found that there were associations between the combined variable and all outcome variables except the prosocial variable. This shows that the combined variable is the strongest predictor of outcomes, which could be attributed to the fact that the combined variable combines traits which themselves can be considered part of well-being. It could also be the case that the combined effect is largely due to anxiety and depression rather than the autistic and ADHD traits. Future research could examine alternative forms of modelling, and it would be beneficial to explore the relationships between autistic, ADHD, anxiety, and depression traits using mediation/moderation analyses. 

The associations between autistic and ADHD traits and well-being and mental health can serve as the basis for making the experience of those with autism and ADHD increasingly holistic, and more focus can be given to coping strategies and management training. This is an important aspect of these conditions because, right from the start, individuals are often given several different diagnoses, and this makes the treatment complicated, which can impact the well-being of individuals. 

Limitations: This study had two major limitations. This study was cross-sectional, so there is no way to infer causality. There was no information collected on previous diagnoses that the students in the study might have received. Future research should, therefore, use a longitudinal design, preferably with an intervention aimed at modifying all traits contributing to the combined risk. It is also important to compare those who have been formally diagnosed with autism, ADHD, anxiety, and depression.

## 5. Conclusions

The present study examined the associations between autistic, ADHD, anxiety, and depressive traits and well-being. The analysis found strong associations between autistic and ADHD traits as well as associations between these traits and mental health traits. The analysis also showed that there were significant effects of the predictor variables on the combined variable consisting of autism, ADHD, well-being, and mental health. This study started by replicating an analysis of the associations between AQ, ADHD, and well-being outcomes carried out with a sample of secondary school students. This study then went on to examine the associations between autistic, ADHD, anxiety, and depression traits and well-being and SDQ outcomes. This study also studied the effect of AQ, ADHD, anxiety, and depression in combination with the formation of a new combined variable and examined the association between this combined variable and well-being outcomes and predictors of well-being. 

This study provided a basis for future research to focus on understanding whether autistic, ADHD, anxiety, and depression traits are interactive and what role they play in well-being. Future research can also find ways to overcome the limitations that were present in the current research and help in the practical management of autism, ADHD, anxiety, and depression, which in turn will increase personal well-being.

## Figures and Tables

**Table 1 healthcare-12-00014-t001:** Descriptive statistics (possible range 1–10 unless indicated).

Variables	N	Mean	Std. Deviation
Anxiety	431	6.19	1.981
Depression	430	4.64	2.120
Positive well-being	427	6.30	1.875
Negative well-being	431	6.16	2.034
Stress	426	6.89	2.010
Social support	429	6.85	1.966
Positive coping	428	6.73	1.829
Negative coping	428	5.92	2.131
Psychological capital	430	5.98	1.824
Work–life balance	427	6.43	2.097
Workload	427	6.47	1.981
Sleepy	424	6.80	2.018
Physical health	426	5.32	1.738
Flow	425	5.75	1.687
Flourishing	425	5.39	1.718
Rumination	427	4.80	2.025
Total ADHD (0–6)	431	2.56	1.755
Total AQ (0–6)	431	3.47	2.075
Conduct (0–10)	431	1.88	1.621
Hyperactivity (0–10)	431	4.66	2.408
Emotional (0–10)	431	5.13	2.498
Peer (0–10)	431	2.48	1.783
Prosocial (0–10)	431	8.05	1.931

**Table 2 healthcare-12-00014-t002:** Correlation between anxiety, depression, autism, and ADHD with outcome variables.

	Anxiety	Depression	Total AQ	Total ADHD
Anxiety	1	0.517 **	0.220 **	0.295 **
Depression	0.517 **	1	0.234 **	0.350 **
Total AQ	0.220 **	0.234 **	1	0.462 **
Total ADHD	0.295 **	0.350 **	0.462 **	1
Conduct	0.101 *	0.157 **	0.197 **	0.231 **
Hyperactivity	0.272 **	0.354 **	0.426 **	0.639 **
Emotional	0.623 **	0.560 **	0.342 **	0.444 **
Peer	0.239 **	0.394 **	0.365 **	0.216 **
Prosocial	−0.034	−0.099 *	−0.093	0.040
Positive well-being	−0.409 **	−0.544 **	−0.189 **	−0.206 **
Negative well-being	0.536 **	0.649 **	0.182 **	0.339 **
Physical health	−0.217 **	−0.238 **	−0.198 **	−0.209 **
Flourishing	−0.348 **	−0.452 **	−0.171 **	−0.254 **

** Correlation is significant at the 0.01 level (2-tailed); * Correlation is significant at the 0.05 level (2-tailed).

**Table 3 healthcare-12-00014-t003:** Correlations between anxiety, depression, AQ, and ADHD and the predictors of well-being.

	Anxiety	Depression	Total AQ	Total ADHD
Anxiety	1	0.517 **	0.220 **	0.295 **
Depression	0.517 **	1	0.234 **	0.350 **
Stressors	0.407 **	0.416 **	0.154 **	0.239 **
Social support	−0.148 **	−0.330 **	−0.237 **	−0.178 **
Positive coping	−0.124 *	−0.274 **	−0.225 **	−0.152 **
Negative coping	0.470 **	0.484 **	0.207 **	0.300 **
Psychological capital	−0.418 **	−0.499 **	−0.238 **	−0.246 **
Work–life balance	0.286 **	0.262 **	0.145 **	0.234 **
Workload	0.269 **	0.188 **	0.125 **	0.204 **
Sleepy	0.312 **	0.336 **	0.162 **	0.339 **
Flow	−0.083	−0.151 **	−0.169 **	−0.294 **
Rumination	−0.059	−0.096 *	−0.060	−0.052

** Correlation is significant at the 0.01 level (2-tailed); * Correlation is significant at the 0.05 level (2-tailed).

**Table 4 healthcare-12-00014-t004:** Regressions with AQ, ADHD, anxiety, and depression and the predictor variables and outcomes.

Outcome	Predictor	Beta	*p*-Value
Positive well-being	Anxiety	−0.173	<0.001
	Depression	−0.452	<0.001
Negative well-being	Anxiety	0.262	<0.001
	Depression	0.484	<0.001
	ADHD	0.110	0.008
Physical health	Depression	−0.132	0.019
	AQ	−0.104	0.047
Flourishing	Anxiety	−0.143	0.005
	Depression	−0.345	<0.001
Conduct	AQ	0.107	0.045
	ADHD	0.156	0.005
Emotional problems	Anxiety	0.414	<0.001
	Depression	0.257	<0.001
	ADHD	0.184	<0.001
	AQ	0.105	0.006
Hyperactivity	Depression	0.127	0.004
	AQ	0.151	<0.001
	ADHD	0.519	<0.001
Peer problems	Depression	0.330	<0.001
	AQ	0.306	<0.001
Prosocial	Depression	−0.127	0.028
	AQ	−0.129	0.018
	ADHD	0.139	0.014

**Table 5 healthcare-12-00014-t005:** Regression analyses with outcome and predictor variables.

Outcome	Predictor	Beta	*p*-Value
Positive well-being	Depression	−0.259	<0.001
	Anxiety	−0.200	0.005
	Social support	0.193	0.004
	Rumination	0.122	0.038
Negative well-being	Anxiety	0.213	<0.001
	Depression	0.471	<0.001
	Stressors	0.190	<0.001
	Rumination	−0.131	0.005
Physical health	Workload	−0.233	0.003
	Rumination	0.129	0.053
Flourishing	Anxiety	−0.153	0.009
	Psychological capital	0.310	<0.001
	Flow	0.335	<0.001
Conduct problems	Depression	0.253	0.004
	Stress	−0.167	0.029
	ADHD	0.168	0.035
	Social support	−0.231	0.002
	Negative coping	−0.268	<0.001
Emotional problems	Anxiety	0.358	<0.001
	Depression	0.162	0.007
	Sleepiness	0.137	0.009
Hyperactivity	AQ	0.193	<0.001
	ADHD	0.425	<0.001
	Flow	−0.150	0.006
Peer problems	Depression	0.333	<0.001
	AQ	0.290	<0.001
	Social support	−0.267	<0.001
Prosocial behaviour	No significant predictors		

**Table 6 healthcare-12-00014-t006:** Correlation between the combined variable and the outcome variables.

	Combined Variable
Combined variable	1
Positive well-being	−0.465 **
Negative well-being	0.574 **
Physical health	−0.304 **
Flourishing	−0.431 **
Conduct	0.239 **
Emotional	0.682 **
Hyperactivity	0.598 **
Prosocial	−0.072
Peer	0.404 **

** Correlation is significant at the 0.01 level (2-tailed).

**Table 7 healthcare-12-00014-t007:** Correlation between the combined variable and the predictor variables.

	Combined Variable
Combined variable	1
Stressors	0.413 **
Social support	−0.323 **
Positive coping	−0.290 **
Negative coping	0.509 **
Psychological capital	−0.478 **
Work–life balance	0.298 **
Workload	0.258 **
Sleepy	0.389 **
Flow	−0.252 **
Rumination	−0.084

** Correlation is significant at the 0.01 level (2-tailed).

**Table 8 healthcare-12-00014-t008:** Regression table of the combined variable and established predictors.

Outcome	Predictor	Beta	*p*-Value
Positive well-being	Combined variable	−0.224	<0.001
	Psychological capital	0.285	<0.001
Negative well-being	Combined variable	0.334	<0.001
	Stressors	0.177	<0.001
Physical health	Combined variable	−0.232	<0.001
Flourishing	Stress	−0.142	<0.001
	Psychological capital	0.364	<0.001
	Flow	0.304	<0.001
Conduct problems	Combined variable	0.331	<0.001
Combined variable	Combined variable	0.405	<0.001
	Negative coping	0.156	<0.001
	Psychological capital	−0.186	<0.001
	Sleepy	0.156	<0.001
Hyperactivity	Combined variable	0.526	<0.001
	Flow	−0.176	<0.001
Peer problems	Combined variable	0.314	<0.001
	Social support	−0.249	<0.001
Prosocial behavior	No significant predictors		

## Data Availability

The data presented in this study are available on request from the corresponding author. The data are not publicly available due to privacy guidelines as the data was collected through the university.

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
