# Peer review of "Associations between Autistic and ADHD Traits and the Well-Being and Mental Health of University Students"

_healthcare, 2023, doi:10.3390/healthcare12010014_

Round 1

Reviewer 1 Report

Comments and Suggestions for Authors

Thank you for entrusting me with the review of the manuscript titled 'Associations between Autistic and ADHD Traits and Well-Being and Mental Health of University Students.' While the authors have chosen an important topic, the manuscript requires further revision for clarity and readability. Herein, I provide my suggestions as follows.

1.     Introduction of the Strengths and Difficulties Questionnaire: In the introduction, when you first mention the Strengths and Difficulties Questionnaire (SDQ), it's important to provide a concise description. The SDQ is a behavioral screening questionnaire developed for children and adolescents. It is used to measure emotional and behavioral difficulties as well as prosocial behavior. This context will help readers understand its relevance to your study.

2.     Elaboration of the "Well-Being Process": Since "well-being process" is a central concept in your study, it's essential to define and discuss it early in the text. Clarify what you mean by this term, perhaps including its components, its significance in the context of mental health, and how it relates to autistic and ADHD traits. This will ensure that readers unfamiliar with the concept can follow your arguments more easily. Importantly, it is challenge to understand what’s negative well-being.

3.     Structural Adjustment Regarding Previous Research: Currently, the focus on your previous research, outlining what has and has not been addressed, narrows its perceived scope. I suggest starting with an overview of the current state of mental health, well-being, and the impact of autistic and ADHD traits in university students, identifying existing literature gaps. Then, link how your previous work aligns with these gaps and describe how your current study seeks to address them, thereby expanding upon or challenging existing knowledge. Concluding the introduction with a clear statement of your study's unique contribution will enhance its readability and emphasize its significance in the wider academic community.

4.     Clarification of the "Combined Effect" Term: Since "combined effect" is crucial in your study, it requires a clearer definition and explanation. This term should be introduced and detailed in the Methods section, not in the Results when you try to explain how you calculate this. Describe how you conceptualize and calculate this combined effect, and reference other studies that have used a similar approach. This will not only clarify your methodology but also place your work within the context of existing research.

5.     Improvement of Readability and Addressing the Knowledge Gap: Enhance the manuscript's readability by clearly stating the knowledge gap your study addresses. Rather than focusing solely on your previous work, highlight how your research contributes to and advances the broader field. Make the narrative more accessible and ensure it communicates the wider implications and novelty of your study.

6.     Clarity in Variable Selection and Regression Analysis: It’s essential to explain the rationale behind your choice of study variables and their inclusion in the regression analysis. Detail why these variables are relevant and how they relate to your research questions. When discussing the regression analysis, clearly describe which variables were used as predictors, their significance, and how they contribute to understanding the relationship between autistic and ADHD traits and student well-being.

Reviewer 2 Report

Comments and Suggestions for Authors

The authors have measured self-reported neurodivergent traits, and wellbeing/mental health related indices in a fairly large sample of University students to examine their associations in this population. The results indicate, in line with existing literature in the autism/ADHD research fields, that there were associations particularly between neurodivergent traits and mental health (i.e., anxiety, depression). There were also correlations between neurodivergent and mental health features with wellbeing and other related features (e.g., behavioural and emotional problems, and some aspects of coping and support, with varying strengths of association across anxiety, depression, autistic and ADHD traits).

For me, the area of strongest potential for the work is the focus on the University context, specifically, and the implications and applications of the findings in that setting.

However, as the manuscript is currently written, it is much more focused on identifying the associations outlined above in neurodivergent groups, which undermines the value of the paper both because the sample used are not an autism/ADHD sample (though some individuals may have a diagnosis, but this is not known), and there is a significant lack of depth provided on existing literature in this area where there are already a range of studies that demonstrate strong associations between autism/ADHD with mental health, and with wellbeing/quality of life, as well as investigating their potential moderating factors in neurodivergent populations. Please see specific feedback on the manuscript below in terms of other areas to develop.

Abstract

·        In the abstract, the authors indicate that the approach is useful to understand the impact on mental health in autism, and the first lines frame the study in terms of research on mental health in autism and ADHD. However, the sample are university students (rather than an autism group) and the analyses are focused on identifying associations between trait level features. Therefore, it feels like too much of a stretch to indicate this will lead to better understanding of the impact on mental health in autism. Rather, it seems the study is addressing associations between variation in mental health and neurodevelopmental features in a general student population, and the importance and novelty of this approach could be made clearer given there is already a growing body of evidence in the literature for associations between autism/ADHD and mental health in neurodivergent populations, specifically.

·         In relation to the SDQ, the authors indicate the influence of ‘outcomes and predictors’, but it does not seem that the design is longitudinal, so this should speak to associations rather than predictive pathways. It’s also unclear what the outcome vs. predictor variables are here (e.g., does this refer to SDQ subscale scores, or something else?).

·         It’s unclear why the authors chose to combine autistic traits, ADHD, anxiety and depression scores into a single variable – what is the value of doing this? Presumably, it’s important to be able to disentangle which features are more or less associated with other features to get at specificity.

·         It would be informative to see some illustrative statistics in the abstract, accompanying the main findings reported.

·         It would also be helpful to understand from the abstract the authors’ views on the conclusions that can/should be drawn from this study and their relevance and importance.

Introduction

·         The authors state that ‘most studies in the autism literature have not used a holistic approach’ – but they do not specify what they mean here (i.e., which kinds of studies in the autism literature, and how is a holistic approach being defined to indicate that it has not been used?) As noted in comments on the Abstract, there is already a growing body of evidence in the literature for associations between autism/ADHD and mental health in neurodivergent populations, specifically, and so this does need clarification.

·         As with comments on the abstract, there is again a general indication that autism and ADHD are captured in their entirety in the approach put forward in this manuscript (in other words, that the research can address questions around associations ‘in autism and ADHD’), where I would advise caution on the phrasing of such points given that the analyses, including of the prior study summarised, are focused at the trait level only.

·         Aspects of the Introduction lack depth, which reduces the potential of the work as a whole. For instance, the authors utilise direct quotations for definitions of concepts/constructs under investigation and there is some over-reliance on existing review papers with relatively few references provided overall to set the scene for the study, even where statements and claims are made (e.g., ‘recent research suggests…” but with no reference provided to where this is drawn from). At times, it is unclear what the authors have done themselves in terms of a literature review, vs. what they are drawing on from previous reviews and meta-analyses in the way some information is currently phrased. A significant amount of space is given to summarising the prevalence rates of mental health in autism, particularly, and it feels that the Introductory section as a whole could be both more focussed on building the foundations for the scope of this specific study itself (I would want to hear more about existing evidence and gaps/limitations in work on University populations and why it’s so important to investigate the proposed associations).

·         The authors mention predictors a lot in the latter part of the Introduction, and based on the prior information provided, presumably here they are referring to SDQ traits etc., however this isn’t specified. Substantial individual predictors for mental health and wellbeing outcomes have been investigated and reported in the literature, so it would be informative to understand which specific ones are the focus of this study (and why).

Methods

·         Is the high proportion of female respondents to the survey reflective of the sex/gender ratio of the wider population this sample was drawn from? This is relevant given literature on sex/gender differences in neurodevelopment and mental health.

·         The authors mention that autism is characterised by traits that are hard to differentiate from other traits, however they do not actually define which are the core traits of autism (and which other traits these may thus be overlapping). Similarly, the information provided about wellbeing seemed quite vague (e.g., phrases like “a wide variety of things are there that come under it”), especially when the authors already included a definition in the Introduction and the Method is a space to operationalise the concepts.

·         I would recommend introducing the measures used, before defining the IV and DV because as the information is currently set out it is unclear to the reader where all these variables are being drawn from and how/why they were selected.

·         The AQ is not a diagnostic questionnaire (e.g., please see http://dx.doi.org/10.1017/S0033291716001082).

·         If it is the case that anxiety and depression traits are being indexed by some items from the WPQ, it would be helpful for the reader to understand what these items encompass in order to evaluate and properly interpret any findings on associations with anxiety/depression, which it doesn’t seem are being separately evaluated using more widely applied clinical/research measures here.

·         In terms of the analytics strategy, the authors should specify what software was utilised to run the analyses, and which specific analyses were run (e.g., Pearson’s correlations, which type of regression approach?).

·         As with earlier comments, I would advise to avoid using terminologies around ‘predictor’ and ‘outcome’ because the study design is cross-sectional and so can only address associations at a single time point. It will also make the definitions of the variables under investigation at each stage of the analyses clearer to follow for the reader.

·         The authors switch back and forth between conceptual language (e.g., autism, ADHD, wellbeing, mental health) and methodological language (e.g., AQ, WPQ, etc.). I would advise consistency, with conceptual language used in narrative sections of the work, and methodological language in the methods/results, again to make it easier for the reader to follow along with exactly what is being assessed when. This is especially important when, as the authors themselves point out, some of the concepts included here are quite broad in the traits that they encompass.

Results

·         In the Methods/Results sections, the authors could be more specific e.g., it is noted there was very little missing data, but how much and what was done to manage it for analyses?

·         How many/what percentage of students were meeting high threshold cutoffs on scales like the AQ, etc.? Is it possible to understand what percentage of the sample also have known neurodevelopmental or mental health diagnoses existing?

·         I’m not sure that the authors need to refer back to their previous study in the middle of the Results, this can be saved to the Discussion so that the reader can here just follow the key findings and begin to develop interpretations. The words saved by doing this would also enable the authors to include key statistical findings in the main text to illustrate the take home points, in addition to these being comprehensively summarised in the tables.

·         Some correlations in Table 2 and 3 overlap and as there are so many variables, it would be helpful for the reader to be given more information here, such as a reminder which measures each of the variables listed in the tables are being drawn from (e.g., SDQ: Conduct), and also more informative subheadings. It’s still difficult to understand how the authors differentiate between what they are referring to as predictors of wellbeing (e.g., they refer to ‘established predictor variable’, but how is this established, do the authors mean in the current study or prior literature?), and outcome variables, firstly because it doesn’t tie in with the design of the study but also because the reader hasn’t yet gotten to the regression model approach, and so it is more helpful to really know exactly which measures (and associated concepts) are being compared here.

·         It’s not clear to me how or why the authors have selected each IV against each DV e.g., some models are including anxiety, depression, ADHD and AQ, while others include just some of these variables.

·         The rationale for creating the combined variable isn’t totally compelling as the correlations between the variables indicated are not consistently high, they are low to moderate. It’s unsurprising that this variable is correlated with mostly all other variables, as it’s collapsing variation across multiple measures into one score. I’m not strongly convinced as to how informative this aspect of the analysis is as the interpretation of the combined variable is not clear.

Discussion

·         Much of the Discussion is dedicated to reiterating the main findings of the study, and so there is little integration of these findings in the context of the wider literature that would be important to emphasise to the reader how this study advances what is already known, how far it agrees (or not) with prior approaches and why, and what these findings might mean in terms of implications and application to impact. For instance, existing literature on associations between mental health and wellbeing/quality of life in neurodivergent groups is available and seems relevant here but is not noted.

·         The authors state that future studies should include an intervention, but what would that be and what would it target? This doesn’t seem to tie in with the rest of the piece as currently written.

Round 2

Reviewer 1 Report

Comments and Suggestions for Authors

The authors have addressed all my questions. Congratulations on their work.